# Design, Synthesis, and Biological Evaluation of 5,6,7,8-Tetrahydrobenzo[4,5]thieno[2,3-*d*]pyrimidines as Microtubule Targeting Agents

**DOI:** 10.3390/molecules27010321

**Published:** 2022-01-05

**Authors:** Farhana Islam, Arpit Doshi, Andrew J. Robles, Tasdique M. Quadery, Xin Zhang, Xilin Zhou, Ernest Hamel, Susan L. Mooberry, Aleem Gangjee

**Affiliations:** 1Division of Medicinal Chemistry, Graduate School of Pharmaceutical Sciences, Duquesne University, 600 Forbes Avenue, Pittsburgh, PA 15282, USA; islamf@duq.edu (F.I.); doshia@duq.edu (A.D.); quaderyt@duq.edu (T.M.Q.); zhangx@duq.edu (X.Z.); zhoux@duq.edu (X.Z.); 2Department of Pharmacology, University of Texas Health Science Center at San Antonio, 7703 Floyd Curl Drive, San Antonio, TX 78229, USA; roblesa3@uthscsa.edu; 3Mays Cancer Center, University of Texas Health Science Center at San Antonio, 7703 Floyd Curl Drive, San Antonio, TX 78229, USA; 4Molecular Pharmacology Branch, Developmental Therapeutics Program, Frederick National Laboratory for Cancer Research, Division of Cancer Treatment and Diagnosis, National Cancer Institute, National Institutes of Health, Frederick, MD 21702, USA; hamele@dc37a.nci.nih.gov

**Keywords:** microtubules, colchicine site, microtubule targeting agents, Gewald reaction

## Abstract

A series of eleven 4-substituted 5,6,7,8-tetrahydrobenzo[4,5]thieno[2,3-*d*]pyrimidines were designed and synthesized and their biological activities were evaluated. Synthesis involved the Gewald reaction to synthesize ethyl 2-amino-4,5,6,7-tetrahydrobenzo[b]thiophene-3-carboxylate ring, and S_N_Ar reactions. Compound **4** was 1.6- and ~7-fold more potent than the lead compound **1** in cell proliferation and microtubule depolymerization assays, respectively. Compounds **4**, **5** and **7** showed the most potent antiproliferative effects (IC_50_ values < 40 nM), while compounds **6**, **8**, **10**, **12** and **13** had lower antiproliferative potencies (IC_50_ values of 53–125 nM). Additionally, compounds **4**–**8**, **10** and **12**–**13** circumvented Pgp and *β*III-tubulin mediated drug resistance, mechanisms that diminish the clinical efficacy of paclitaxel (PTX). In the NCI-60 cell line panel, compound **4** exhibited an average GI_50_ of ~10 nM in the 40 most sensitive cell lines. Compound **4** demonstrated statistically significant antitumor effects in a murine MDA-MB-435 xenograft model.

## 1. Introduction

Microtubules have long been recognized as effective targets for the treatment of many human malignancies [1,2]. Microtubules are involved in a variety of cellular functions including mitosis, motility, intracellular transport, trafficking and organization, including positioning of organelles [1,3,4]. Molecules binding to tubulin and interrupting tubulin dynamics are recognized as microtubule targeting agents (MTAs), and they have been used clinically as single agents or in combinatorial regimens for the effective treatment of leukemia, lymphoma and various solid tumors [2,3,5]. MTAs are a highly diverse class of cytotoxic agents that include a variety of different chemical scaffolds (Figure 1) [2,3,6].

MTAs can be classified into two major groups: (1) microtubule destabilizers that initiate microtubule depolymerization; and (2) microtubule stabilizers that promote the polymerization of tubulin into microtubules [1,2]. Additionally, MTAs are further divided into seven groups based on their binding sites [6,7]. Two binding sites on tubulin/microtubules have been identified for microtubule stabilizers [8]. First, the taxane site is located on the interior of the microtubule, and all clinically approved microtubule stabilizers, including paclitaxel (Figure 1), docetaxel (Figure 1), cabazitaxel and ixabepilone, bind to this site [8]. The taccalonolides (Figure 1), zampanolide and cyclostreptin are compounds that bind covalently within the taxane site, but to date have not been evaluated clinically [8,9,10]. The second stabilizer site is the laulimalide/peloruside site, which is located on the exterior of the microtubule and named for the natural products that bind to this site [8]. The clinical development of compounds binding to this site has been limited by a lack of in vivo efficacy for laulimalide [11] and supply challenges for peloruside A [12]. In the class of microtubule destabilizers, five sites have been identified: the vinca site, the colchicine site (CS), the maytansine site, the pironetin site [6], and more recently the gatorbulin site (gatorbulin-1, Figure 1) defined by the cyclic peptide of the same name [7]. The vinca alkaloids vinblastine, vincristine, vindesine, and vinorelbine, as well as other structurally unique/unrelated compounds, including eribulin (Figure 1), bind within the vinca site, which is located at the interdimer interface between two tubulin heterodimers in a protofilament [6]. The colchicine binding site is located on *β*-tubulin at the intradimer interface of the *αβ*-tubulin heterodimer [6]. The maytansine site is also on *β*-tubulin, in close proximity but nonoverlapping with the vinca site [13]. The gatorbulin site is located at the intradimer interface of tubulin similarly to the vinca alkaloids, but binding within this site is differentiated by extensive contacts with *α*-tubulin [7]. The pironetin site is the only MTA localized exclusively on *α*-tubulin [14].

While the clinically useful vinca alkaloids vinblastine and vincristine were approved decades ago, new MTAs have been approved more recently for clinical use. Eribulin (Figure 1), a simplified synthetic analogue of the natural product halichondrin B, is a microtubule depolymerizer that has unique properties [5,15] and significant utility in the treatment of advanced breast cancer [15]. The dolastatin 10 analogue monomethyl auristatin E and maytansine (Figure 1) analogues are employed as the cytotoxic payloads of antibody-drug conjugates (ADCs) that have found clinical utility [16]. These unconjugated MTAs were too toxic for systemic administration, but their antibody-directed delivery to cancers was designed to reduce off-target toxicities [16]. Continuing challenges with clinically approved MTAs, including the taxanes and vinca alkaloids, are the incidence of dose-limiting side effects and limited efficacy due to multidrug resistance [17]. Cancer cells and patients demonstrate resistance to clinically used agents as a result of the expression of the drug efflux pump P-glycoprotein (Pgp) and the *β*III-tubulin isotype [2,18,19], leading to efforts to identify new MTAs that can overcome these mechanisms of drug resistance.

CS inhibitors, including derivatives of combretastatin A-4 (CA-4, Figure 1), have been extensively studied, and several have been evaluated in clinical trials, including combretastatin A-4 phosphate (CA-4P/fosbretabulin), the combretastatin CA-1P prodrug (OXi4503), 2-methoxyestradiol, AVE8062, CKD-516, BNC105P, ABT-751, CYT-997, ZD6126, plinabulin (NPI-2358) and MN-029 [20,21,22]. Colchicine itself is approved for the treatment of gout but is not employed as an anticancer agent due to toxic side effects at the doses necessary for efficacy [23]. However, other CS agents have exhibited promising potential as anticancer candidates [20,21,22]. Many compounds that interact with the CS are able to overcome multiple mechanisms of drug resistance [24]. This suggests that the development of MTAs targeting the CS has the potential to overcome limitations associated with existing drugs and perhaps improve clinical outcomes. This has been challenging, however, and to date no CS agent has received FDA approval for anticancer indications [22]. There is an urgent need to develop new tubulin inhibitors with fewer side effects and good oral bioavailability that are less prone to clinically relevant drug resistance mechanisms.

## 2. Results and Discussion

### 2.1. Rationale

We previously reported [25,26] N4-substituted-pyrimido[4,5-b]indole-4-amines (**1**–**3**) (Figure 2) with potent microtubule depolymerization activity, with EC_50_ values of 130, 1100 and 1200 nM for the lead compounds **1**, **2** and **3**, respectively. In MDA-MB-435 human cancer cells, the IC_50_ values for antiproliferative effects for compounds **1**, **2** and **3** were 14.7, 89.1 and 130 nM, respectively. This study exploits four medicinal chemistry strategies for the design of potent MTAs based on the lead compounds **1**–**3**. The strategies are (1) isosteric replacement; (2) decrease numbers of sp2 bonds; (3) variation of substitutions at the 2-position; and (4) conformational restriction.
(1)Isosteric replacement: To explore the activities of compounds with the 4,5,6,7-tetrahydrobenzo thiophene scaffold on both inhibition of cancer cell proliferation and microtubule depolymerization, we carried out the isosteric replacement of the scaffold -NH- of the lead compounds **1**–**3** by sulfur (-S-) to afford target compounds **4**–**14** (Table 1). Isosteric replacement of -NH with (-S-) has literature precedence in improving antiproliferative and microtubule depolymerizing activities [27]. Moreover, pharmacological applications of 5,6,7,8-tetrahydrobenzo[4,5]thieno[2,3-*d*]pyrimidines have been extensively illustrated in various reports in the literature [28,29,30,31,32,33,34,35,36,37,38]. In addition, the lead tricyclic compounds and the proposed target compounds incorporate a p-methoxyphenyl substitution akin to colchicine and CA-4 (Figure 1). The nature of the heteroatom substitution (S for NH) affects hydrogen bond (HB) strength [39]. Thus, it was also of interest to isosterically replace the oxygen atom of the 4′-OCH_3_ of **4**, **8** and **9** with a sulfur moiety to afford **5**, **10**, and **11**, in analogy to **2**.(2)Decrease numbers of sp2 bonds: Drug candidates show a higher clinical success rate with one or more sp3 hybridized carbon atoms as compared to “flat” molecules, due to low aqueous solubility of purely aromatic compounds [40]. One of the major limitation of some MTAs, particularly the taxanes, is their poor water solubility [41]. Thus, water-soluble MTAs are highly coveted, and an enormous effort continues to chemically modify and/or formulate analogues to increase their water solubility. Increasing ‘aromatic proportion’ in a molecule has a detrimental effect on the solubility [40]. The fraction of sp3 hybridized carbon atoms (Fsp3), in other words, the fraction of carbon atoms that are saturated, correlates positively with water solubility [40]. In an attempt to both increase the water solubility as well to probe the potential interactions with the hydrophobic pocket in the CS, we designed target compounds **4**–**14** by incorporating sp3 hybridized carbon atoms in the tricyclic scaffold of the lead compounds **1**–**3**.(3)Variation of the substituents at the 2-position: Compound **7** was specifically designed to determine the effect of replacing the 2-NH_2_ in **4** with a 2-H. This allows an exploration of the 2-NH_2_ and hydrogen bond interactions with corresponding amino acids at the CS. It was also of our interest to observe the effect of isosteric replacement of 2-NH_2_ on compound **4** with a 2-CH_3_ to afford **8**. This would also provide information regarding the activity on the replacement of H with CH_3_ at the 2-position in the tricyclic scaffold.(4)Conformational restriction: Conformational restriction or rigidification of a ligand can decrease the entropic penalty [42]. The ligand can adopt a preferred conformation for binding, which might lead to enhanced potency for a given physiological target [42]. In an effort to better define the conformational requirements for biological activities, we systematically incorporated various groups to restrict bond rotations. The conformation of **9** (Figure 3) is determined by three rotatable single bonds: the 4-position C-N bond (bond a), the 1′-position C-N bond (bond b) and the 4′-position C-O bond (bond c). Conformational analysis via molecular modeling and ^1^H NMR studies [25] suggest that the methyl group on the aniline nitrogen in **1** restricted the free rotation of bond a as well as bond b (Figure 2) and consequently restricted the conformation of the anilino ring. To study the significance of conformational restriction on biological activities, we first designed compounds **8** and **9**. In **9,** the rotation of bonds a and b was restricted by incorporating a methyl group at the N4- position to afford compound **8**. Incorporation of tetrahydroquinoline rings in **6** and **12** further restricted bond b of **4** and **8**. The design of compound **13** via the incorporation of a 5-methoxy naphthalene ring provided a further element of conformational restriction.

### 2.2. Molecular Modeling

Computational modeling studies were performed to elucidate the binding mode of the lead and target compounds **1**–**14** and probe the possible interactions with the CS. Compounds **1**–**14** were docked into the CS (PDB: 6BS2, 2.65 Å) [43] of tubulin using Maestro, Schrödinger 2020-2, New York, NY, USA [44]. Figure 4a shows the docked pose of **4** (cyan) superimposed with colchicine (pink) in the X-ray crystal structure of the CS [43]. Multiple low-energy conformations (within −9.68 to −10.89 kcal/mol) were obtained on docking. The pyrimidine and cyclohexene rings of the tricyclic 4,5,6,7-tetrahydrobenzo thieno[2,3-*d*]pyrimidine scaffold of **4**–**14** (compound **4**, Figure 4b is a representative of **5**–**14**) overlapped with the A- and C- rings of the colchicine, respectively. The concave structure created by the 4,5,6,7-tetrahydrobenzo thieno[2,3-*d*]pyrimidine scaffold of **4**–**14** overlapped well with the B- ring of colchicine. The N1 of the pyrimidine and the 2-NH_2_ of compounds **4**–**6** formed water-mediated hydrogen bond interactions with the backbone of Cysβ239. In compounds **8**–**14,** the 2-CH_3_ interacted with the hydrophobic amino acids Valβ236 and Ileβ316. The cyclohexene ring of **4**–**14** formed hydrophobic interactions with Leuβ246, Alaβ248 and Metβ257. The N4-Me moiety of **4**, **5**, **7**, **8**, **10** and **13** formed hydrophobic interactions with Alaβ314 and Alaβ352. The 4′-OMe-Ph of **4**, **6**–**9** and **12**–**14** is oriented towards the pocket formed by polar amino acid residues Lysβ350, Thrβ312 and Asnβ256. The docked score of compound **4** was −10.89 kcal/mol, and for compounds **5**–**14**, the docked scores were in a range of −9.68 to −10.72 kcal/mol (Appendix A Appendix A). The pyrimidine ring of **4** overlaps with the pyrimidine ring of the crystalized ligand of 6BS2 (Appendix A Appendix A).

### 2.3. Chemistry

Compounds **4**–**14** were synthesized according to the synthetic routes outlined in Figure 1, Figure 2, Figure 3 and Figure 4. The Gewald reaction (Figure 1) was carried out on a solution of sulfur in ethanol, to which cyclohexanone **15** and ethyl cyanoacetate were added. Morpholine was added dropwise to the solution to obtain **16**. Cyclization of **16** with chloro-formamidine hydrochloride, formamide and acetonitrile afforded **17**, **18** and **19**, respectively, using reported methods [25,45,46]. Chlorination [47] of **17**–**19** with POCl_3_ and pyridine in toluene afforded **20**–**22** in 68–75% yield.

The monomethylated aniline, 5-methoxy-*N*-methylnaphthalen-2-amine (**26**), was synthesized over three steps from 6-aminonaphthalen-1-ol (**23**) (Figure 2) [48]. Boc protection of **23** in THF at r.t. yielded tert-butyl(5-hydroxyaphthalen-2-yl)carbamate (**24**) in 84% yield, followed by methylation with methyl iodide and sodium hydride to provide **25**. Finally, Boc deprotection [49] using trifluoracetic acid (TFA) afforded 5-methoxy-*N*-methylnaphthalen-2-amine (**26**) in 91% yield.

Intermediates **20**–**22** were subjected to S_N_Ar reactions using appropriate anilines in isopropanol or toluene to afford final compounds **4**–**7**, **9**, **11**, **12** and **14** (Figure 3). Compounds **9**, **11** and **14** were dissolved in DMF, followed by portion-wise addition of sodium hydride and iodomethane. Reaction mixtures were stirred for 2 h at r.t. to afford final compounds **8**, **10** and **13**, respectively (Figure 4).

### 2.4. Biological Evaluations and Discussion

#### 2.4.1. Antiproliferative and Microtubule Depolymerization Effects

We investigated the microtubule depolymerization and the antiproliferative activities of compounds **4**–**14** (Table 2). At a concentration of 10 µM, the compounds that caused at least 50% microtubule depolymerization were further evaluated to determine their EC_50_ values, the concentration that causes the loss of 50% of cellular microtubules as visualized microscopically. Compounds that caused microtubule depolymerization at 10 μM were further evaluated for antiproliferative potency in the drug-sensitive MDA-MB-435 cancer cell line, and the IC_50_ (concentration required to cause 50% inhibition of proliferation) values were determined using the sulforhodamine B assay (SRB assay). Compound **4**, the 2-NH_2_ analogue of 5,6,7,8-tetrahydrobenzo[4,5]thieno[2,3-*d*]pyrimidine, was the most potent compound of this series for both microtubule depolymerizing and antiproliferative effects, with an EC_50_ of 19 nM and an IC_50_ of 9.0 nM (Table 2). Compound **4** was 7-fold more potent than **1** for microtubule depolymerizing effects, indicating that the 5,6,7,8-tetrahydrobenzo[4,5]thieno[2,3-*d*]pyrimidine ring is significantly better for microtubule depolymerizing activity than the pyrimido[4,5-b]indole ring of **1** (Table 2) and that it additionally contributes to improvements in antiproliferative potency. We next evaluated the importance of the 4′-OMe group of compound **4** by replacing it with an isosteric 4′-SMe (**5**). The resulting compound **5** was 15-fold more potent than the corresponding lead compound **2** with respect to microtubule depolymerization activity and was additionally 2.3-fold more potent for antiproliferative effects compared to **2**. Clearly, this further substantiated the importance of an S in the scaffold over an NH. However, comparing compounds **4** and **5** indicated that the 4′-OMe was better than the 4′-SMe. Compound **6**, the tetrahydroquinoline-substituted compound, a conformationally restricted analogue of **4** around bond b, was 6-fold less potent for antiproliferative effects and for microtubule depolymerizing effects than **4**, indicating that conformational restriction in **6** is detrimental to these biological activities.

We next focused on substituting the 2-position of the 5,6,7,8-tetrahydrobenzo[4,5]thieno[2,3-*d*]pyrimidine. The corresponding 2-H analogue **7** of lead compound **3** displayed ~27-fold increased potency in the microtubule depolymerizing assay as compared with **3** (Table 2), indicating a more effective engagement with tubulin. Compound **7** was 3.5-fold more potent than compound **3** for antiproliferative effects. The 2-Me analogue **8** displayed slightly less potency compared to the 2-H compound **7** for both microtubule depolymerizing and antiproliferative effects, yet it had 2.7- and 5.8-fold lower potency than the 2-NH_2_ compound **4** in the microtubule depolymerization assay and antiproliferative assay, respectively. Compound **8** however, with a 2-Me, had a lower EC_50_/IC_50_ ratio (1.0 for **8** as compared to 2.2 for **4**), indicating a tighter correlation between the microtubule depolymerizing effects and the cancer cell cytotoxicity. Compound **10** with a 4′-SMe group and a 2-Me analogue of compound **5** had 2-fold lower potency than **5** in both assays.

In compounds **12** and **13**, conformational restrictions of the N4-phenyl moiety of **8** about bond b with a 1,2,3,4-tetrahydroquinoline moiety and 5′-methoxy naphthalene, respectively, caused a 2.3–3-fold decrease in potency for **12** in antiproliferative and microtubule depolymerizing effects compared to **8** and a 1.5- to 2-fold drop in potency for **13** as compared to **8**. Compounds **9**, **11**, and **14** did not show any activity in the microtubule depolymerization assay, and these were not evaluated for antiproliferative effects, corroborating our previous reports that the N4-Me is crucial for MT activity [26].

For compounds **4**–**8**, **10**, **12** and **13,** the EC_50_/IC_50_ ratios (Table 2) ranged from 1 to 2.2, which is better than the EC_50_/IC_50_ ratios for lead compounds **1**–**3** (8.8, 12 and 9.2 for lead compounds **1**, **2** and **3**, respectively). These lower values for compounds **4**–**8**, **10**, **12** and **13** suggest a cytotoxic mechanism of action that is primarily microtubule-dependent.

#### 2.4.2. Inhibition of Tubulin Assembly and Colchicine Binding

Compounds **4**, **5**, **7**, **8** and **10** were evaluated for their direct effects on purified tubulin assembly and for inhibition of colchicine binding (Table 3). Compounds **4**, **5**, **7**, **8** and **10** inhibited tubulin assembly with activities better than those of the lead compounds **1**–**3** as well as CA-4. Compound **4** was 2-fold more potent than the lead **1** as an inhibitor of tubulin assembly. On the other hand, compounds **5**, **7**, **8** and **10** were 2-fold more potent than the standard CA-4. Moreover, compounds **5** and **7** were 5-fold more potent as inhibitors of tubulin assembly than the corresponding lead compounds **2** and **3**, respectively. Compounds **4**, **5**, **7**, **8** and **10** inhibited the binding of [^3^H]colchicine to tubulin by 89–99%, whereas the lead compounds **1**, **2** and **3** showed 84, 67, and 62% inhibition of [^3^H]colchicine binding, respectively. Thus **4**, **5**, **7**, **8** and **10** were more active than the initial lead compounds **1**–**3**. These results clearly demonstrated that these compounds are CS MTAs.

#### 2.4.3. Effect on βIII-Tubulin and Pgp-Mediated Cancer Cell Resistance

Compounds **4**–**8**, **10**, **12**, and **13** were evaluated for their abilities to overcome βIII-tubulin mediated drug resistance using an isogenic HeLa cell line pair (Table 4). Consistent with the results obtained in MDA-MB-535 cells, compound **4** was the most potent in the series in the HeLa and HeLa WT βIII cell lines, with 1.6-fold higher potency than the lead compound **1**. Compounds **5** and **7** showed 2-fold higher potency than the lead compounds **2** and **3** in HeLa and HeLa WT βIII cell lines. The Rr values (Table 4) were calculated by dividing the IC_50_ of the βIII-tubulin expressing line by the IC_50_ obtained in the parental HeLa cells. The expression of βIII-tubulin is known to lead to paclitaxel resistance, and paclitaxel has an Rr value of 8.6 in this cell line pair (Table 4). The target compounds **3**–**8**, **10**, **12**, and **13** have Rr values ~1.0 (Table 4), suggesting that they circumvent βIII-tubulin mediated drug resistance, in contrast to paclitaxel.

The potent MTAs **4**–**8**, **10**, **12**, and **13** were also evaluated for their activity in the SK-OV-3 ovarian carcinoma cell line and the Pgp-expressing subline SK-OV-3 MDR1-M6/6 (Table 4). In these cell lines, compound **4** was again the most potent compound in the series. Comparison of the IC_50_ values in the parental SK-OV-3 and genetically manipulated SK-OV-3 MDR1-M6/6 cell line allows for the calculation of a relative resistance value, designated Rr. This value is calculated by dividing the IC_50_ value obtained in the Pgp-expressing SK-OV-3 MDR1-M6/6 cells by the IC_50_ obtained in the parental SK-OV-3 cells. Paclitaxel, a known Pgp substrate, has an Rr value of 240, while CA-4, a poor Pgp substrate, has an Rr value of 1.3 (Table 4). Compound **4** had IC_50_ values in SK-OV-3 and SK-OV-3 MDR1-M6/6 cells comparable to that of CA-4 and an Rr of 1.5, indicating that it is able to overcome drug resistance mediated by Pgp. Here, a correlation between a cell-based assay and a biochemical assay is not always observed, which might be due in part to the ability of the compounds to cross the cell membrane and accumulate intracellularly. Compounds **5**–**8**, **10**, **12**, and **13** also had Rr values ≤ 1.5, suggesting that they are all poor substrates for Pgp-mediated transport and have advantages over the taxanes and vinca alkaloids in multidrug-resistant cancer cells.

#### 2.4.4. Activity of Compound **4** in the NCI Cancer Cell Line Panel

Compound **4**, the most potent compound of the series, was selected for evaluation in the NCI-60 cancer cell line panel [50], and it had a GI_50_ (concentration causing 50% inhibition of cell proliferation) of ~10 nM against 40 of the 60 cancer cell lines (Table 5). Compound **4** had better potency than the lead compound **1** in 50 cancer cell lines. (better by 5 to 6-fold in leukemia, 2 to 17-fold in NSCLC, 2 to 6-fold in colon cancer, 2 to 5-fold in CNS cancer, 2 to 25-fold in melanoma, 2 to 5-fold in ovarian cancer, 2 to 9-fold in renal cancer, 2 to 5-fold in prostate cancer, and 2 to 6-fold in breast cancer compared to lead compound **1**) [26]. Thus **4**, the thiophene-fused analogue, is up to 25-fold more potent than our previously published lead [26].

#### 2.4.5. Antitumor Activity of Compound **4** in MDA-MB-435 Xenografts

Compound **4** was selected for further evaluation in an in vivo xenograft mouse study in light of its nanomolar potency in vitro in the NCI-60 cancer cell line panel and its potent microtubule depolymerization activity. The in vivo effects of **4** were tested in the MDA-MB-435 xenograft model (Figure 5). After conducting initial dose tolerance testing, **4** was administered at a dose of 75 mg/kg 3 × a week where it caused moderate weight loss yet had statistically significant antitumor effects as compared to the control at day 14, the end of the trial. In this trial, there was a trend toward antitumor effects with paclitaxel (15 mg/kg), but this did not reach statistical significance at any day or at trial conclusion.

## 3. Materials and Methods

### 3.1. Chemistry

All evaporations were carried out under a vacuum using a rotary evaporator. Analytical samples were dried in vacuo (0.2 mmHg) in a CHEM-DRY drying apparatus over P_2_O_5_ at 50 °C. Thin-layer chromatography (TLC) was performed on Whatman Sil G/UV254 silica gel plates (Whatman International Ltd., Maidstone, England), and the spots were visualized by irradiation at 254 nm. Proportions of solvents used for TLC are by volume. All analytical samples were homogeneous on TLC in at least two different solvent systems. Column chromatography was performed on a 70–230 mesh silica gel (Fisher Scientific, Waltham, MA, USA) column. The amount (weight) of silica gel for column chromatography was in the range of 50–100 times the amount (weight) of the crude compounds being separated. Columns were wet-packed with appropriate solvent unless specified otherwise. Melting points were determined using a digital MEL-TEMP II melting point apparatus with FLUKE 51 K/J electronic thermometer or using an MPA100 OptiMelt (Stanford Research Systems, Sunnyvale, CA, USA) automated melting point system and are uncorrected. Nuclear magnetic resonance spectra for protons (^1^H NMR) were recorded on Bruker Avance II 400 (Billerica, MA, USA) (400 MHz) and 500 (500 MHz) systems and were analyzed using MestReC NMR (Mestrelab research, San Diego, CA, USA, data processing software. The chemical shift (δ) values are expressed in ppm (parts per million) relative to tetramethylsilane as an internal standard: s, singlet; d, doublet; t, triplet; q, quartet; m, multiplet; br, broad singlet; exch, protons exchangeable by addition of D_2_O.

Elemental analyses or high-performance liquid chromatography (HPLC)/mass analysis were used to determine the purities of the target compounds. Elemental analyses were performed by Atlantic Microlab, Inc., Norcross, GA, USA. Elemental compositions are within ±0.4% of the calculated values and indicate >95% purity. Fractional moles of water or organic solvents found in some analytical samples could not be removed despite 24–48 h of drying in vacuo and were confirmed where possible by their presence in the ^1^H NMR spectra. Mass spectral data were acquired on an Agilent G6220AA TOF LC/MS system using the nano ESI (Agilent chip tube system with infusion chip). HPLC analysis was performed on a Waters HPLC system using a XSelect CSH C18 column. Peak area of the major peak versus other peaks was used to determine purity. All solvents and chemicals were purchased from Sigma-Aldrich Co, USA. or Fisher Scientific Inc, USA. and were used as received.

Ethyl 2-amino-4,5,6,7-tetrahydrobenzo[b]thiophene-3-carboxylate (**16**): 4-Cyclohexanone **15** (5.27 mL, 50.95 mmol) and morpholine (4.39 mL, 50.95 mmol) were added to a mixture of ethyl cyanoacetate (5.42 mL, 50.95 mmol) and sulfur (13.07 g, 50.95 mmol) in ethanol (25 mL). The mixture was stirred at room temperature for 1 h, then at 60 °C for 12 h. The reaction mixture was cooled to room temperature, and the solvent was removed in vacuo. The crude product was purified by flash column chromatography on a silica column using hexane/ethyl acetate (10:1) as eluent to obtain compound **16** (7.80 g, 34.64 mmol, 68% yield) as a light-yellow solid. TLC Rf = 0.67 (hexane: EtOAc, 3:1); mp, 194–195.7 °C (lit. [51]192–193 °C); ^1^H NMR (400 MHz, DMSO-d_6_) δ 6.66 (s, 2H, br, exch., NH_2_), 4.36 (q, J = 7.0 Hz, 2H, -CH_2_CH_3_), 2.60 (m, 2H, -CH_2_), 2.41 (m, 2H, -CH_2_), 1.68–1.65 (m, 4H, -CH_2_), 1.42 (t, J = 7.0 Hz, 3H, -CH_2_CH_3_). The ^1^H-NMR matches the ^1^H-NMR of the reported compound in the literature [51]. This compound was used for the next reaction without further characterization.

2-Amino-5,6,7,8-tetrahydrobenzo[4,5]thieno[2,3-*d*]pyrimidin-4(3H)-one (**17**): Methyl sulfone (15 g), intermediate **16** (5.0 g, 22.19 mmol) and chloroformamidine hydrochloride (5.10 g, 44.38 mmol) were mixed in a round bottom flask. The reaction mixture was stirred at 140 °C for 4 h. The reaction was quenched with 100 mL water, cooled in an ice bath and basified to pH 8.0 using an aqueous NH_4_OH solution. The precipitate was collected by filtration, dried (using Na_2_SO_4_) and afforded 3.24 g (66%) of **17** as a brown solid. TLC Rf = 0.60 (CHCl_3_: MeOH, 5:1). The product **17** was not purified further and taken to the next step without characterization.

4-Chloro-5,6,7,8-tetrahydrobenzo[4,5]thieno[2,3-*d*]pyrimidin-2-amine (**20**): Compound **17** (1.96 g, 8.84 mmol) was chlorinated with phosphorus oxychloride (0.80 mL, 8.84 mmol) and pyridine (0.7 mL, 8.84 mmol) in toluene (15 mL). The reaction was kept at reflux for 4 h. The POCl_3_ was evaporated, and the mixture was cooled in an ice bath. The mixture was neutralized using an aqueous NH_4_OH solution to yield a precipitate. The precipitate was collected by filtration, washed with water, dried and dissolved in MeOH. To the solution was added silica gel (1 g), and the solvent was removed under reduced pressure to provide a silica gel plug. Column chromatography was performed with hexane and ethyl acetate (10:1) to generate **20** (2.0 g, 6.19 mmol, 70%) as a brown solid. TLC Rf = 0.68 (hexane: EtOAc, 3:1); mp 234 °C; ^1^H NMR (400 MHz, DMSO-d_6_): δ 5.28 (s, br, 2H, exch., 2-NH_2_), 2.74 (t, 2H, -CH_2_), 1.92 (t, 2H, -CH_2_), 1.56–1.50 (m, 2H, -CH_2_), 1.43–1.38 (m, 2H, -CH_2_). This compound was used for the next reaction without further characterization.

4-Chloro-5,6,7,8-tetrahydrobenzo[4,5]thieno[2,3-*d*]pyrimidine (**21**): Treatment of **16** (5.0 g, 22.19 mmol) with formamide (4.42 mL, 110.96 mmol) was carried out in a microwave vessel at 180 °C for 12 h. The reaction was cooled to room temperature, and 50 mL water was added to the mixture. The precipitate was collected and dried under high vacuum to afford **18** as a white solid in 72% yield (3.30 g). The product **18** was taken to the next step without characterization. Chlorination of **18** (3.0 g, 14.54 mmol) was performed using POCl_3_ (1.4 mL, 14.54 mmol) and pyridine (1.17 mL, 14.54 mmol), and the mixture was kept at reflux for 8 h. The solvent was removed by evaporation, and the residue was neutralized with ammonia in water solution to generate a pale-yellow precipitate. The precipitate was collected by filtration. To the precipitate was added methanol and 2.0 g of silica gel. The solvent was removed under reduced pressure, and a silica plug was prepared. A flash column chromatographic separation was performed using ethyl acetate-hexane as eluent to afford 1.96 g of **21** (8.73 mmol, 60%) as a pale-yellow solid. TLC Rf = 0.77 (Hexane: EtOAc, 3:1); mp, 210–112 °C; ^1^H NMR (400 Hz) (Me_2_SO-d_6_) δ 8.51 (s, 1H, Ar), 2.86–2.82 (m, 2H, -CH_2_), 2.78–2.74 (m, 2H, -CH_2_), 1.84–1.78 (m, 4H, -CH_2_CH_2_). Anal. Calcd. for C_10_H_9_ClN_2_S: C, 53.45; H, 4.04; Cl, 15.78, N, 12.47; S, 14.27. Found: C, 53.58; H, 4.07; Cl, 15.57, N, 12.37; S, 14.12.

2-Methyl-5,6,7,8-tetrahydrobenzo[4,5]thieno[2,3-*d*]pyrimidin-4(3H)-one (**19**): Compound **16** (2.5 g, 11.10 mmol) was dissolved in 20 mL of acetonitrile and hydrogen chloride gas was bubbled through for 30 min. The mixture was stirred at room temperature overnight. The residue was dissolved in 10 mL distilled water and treated with ammonia in water solution to generate a white precipitate. The precipitate was collected by filtration to afford 1.27 g (5.77 mmol, 52%) of **19** as a white solid. TLC Rf = 0.11 (hexane: EtOAc, 3:1); mp >250 °C (lit. [51] 285 °C) ^1^H NMR (400 MHz, DMSO-d6): δ 12.10 (s, 1H, exch., -NH), 3.02–2.95 (m, 2H, -CH_2_-), 2.87–2.84 (m, 2H, -CH_2_-), 2.66 (s, 3H, -CH_3_), 1.84 (t, J = 3.1 Hz, 4H, -CH_2_-). This compound was used for the next reaction without further characterization.

4-Chloro-2-methyl-5,6,7,8-tetrahydrobenzo[4,5]thieno[2,3-*d*]pyrimidine (**22**): Chlorination of **18** (1.0 g, 4.54 mmol) was carried out using POCl_3_ (0.63 mL, 6.81 mmol) and pyridine (0.54 mL, 6.81 mmol) in 15 mL of xylene under reflux for 6 h. The solvent was evaporated and neutralized with ammonia in water solution to generate a light-yellow precipitate. The precipitate was collected by filtration to afford **22** as a light-yellow solid (704.42 mg, 2.95 mmol, 65%). TLC Rf = 0.83 (hexane: EtOAc, 3:1); mp, 228–230 °C; ^1^H NMR (400 Hz) (Me_2_SO-d_6_) δ 2.79–2.85 (m, 2H, -CH_2_-), 2.84–2.80 (m, 2H, -CH_2_-), 2.58 (s, 3H, -CH_3_), 1.81–1.77 (m, 4H, -CH_2_-). This compound was used for the next reaction without further characterization.

#### General Procedure for Synthesis of 4–14

Compounds **20**–**22** were dissolved in isopropanol, followed by addition of 1–2 drops of HCl and the appropriate anilines. The reaction mixture was stirred for 4–8 h at reflux. The reaction mixture was cooled, and silica gel was added to the solvent mixture to prepare a silica gel plug. A flash column chromatographic separation was performed using ethyl acetate-hexane as eluent to afford **4**–**7**, **9**, **11**, **12** and **14** with yields of 48–68%. Compounds **9**, **11** and **14** were added to NaH in DMF with drop-wise addition of iodomethane to obtain **8**, **10** and **13**, respectively, in 57–70% yield.

N4-(4-methoxyphenyl)-N4-methyl-5,6,7,8-tetrahydrobenzo[4,5]thieno[2,3-*d*]pyrimidine-2,4-diamine (**4**): To a solution of **20** (250 mg, 1.04 mmol) in isopropanol (20 mL), 1–2 drops of HCl were added, followed by addition of 4-methoxy-*N*-methylaniline (157.3 mg, 1.15 mmol), followed by reflux for 6 h. The reaction mixture was cooled to room temperature, silica gel (500 mg) was added, and the solvent was removed under reduced pressure. Purification was performed by column chromatography using 1% MeOH in CHCl_3_ as the eluant, and fractions containing the product (TLC) were pooled. The solvent was evaporated to give a white solid that was washed with CHCl_3_ to afford 230.0 mg (65% yield) of **4**. TLC Rf = 0.40 (CHCl_3_: MeOH, 20:1); mp 167–168.1 °C; ^1^H NMR (400 MHz, DMSO-d_6_) δ 6.93 (d, J = 9.0 Hz, 2H, Ar), 6.82 (d, J = 9.0 Hz, 2H, Ar), 5.19 (s, br, 2H, exch., 2-NH_2_), 3.81 (s, 3H, -OCH_3_), 3.46 (s, 3H, N_4_-CH_3_), 2.67–2.63 (m, 2H, -CH_2_), 1.66–1.60 (m, 4H, -CH_2_CH_2_), 1.49–1.45 (m, 2H, -CH_2_). Anal. Calcd. for C_18_H_20_N_4_OS 0.29 H_2_O: C, 62.54; H, 6.00; N, 16.20; S, 9.27. Found: C, 62.58; H, 6.03; N, 16.10; S, 9.21.

N4-methyl-N4-(4-(methylthio)phenyl)-5,6,7,8-tetrahydrobenzo[4,5]thieno[2,3-*d*] pyrimidine-2,4-diamine (**5**): To a solution of **20** (150 mg, 0.625 mmol) in toluene (8 mL), 1–2 drops of HCl were added, followed by addition of *N*-methyl-4-(methylthio)aniline (105.5 mg, 0.688 mmol), and the mixture was kept under reflux for 6 h. The reaction mixture was cooled to room temperature, silica gel (500 mg) was added, and the solvent was removed under reduced pressure. Purification was performed by column chromatography using 1% MeOH in CHCl_3_ as the eluant, and the fractions containing the product (TLC) were pooled. The solvent was evaporated to give a pale-yellow solid that was washed with CHCl_3_ to afford 109.0 mg (49% yield) of **5**. TLC Rf = 0.35 (CHCl_3_: MeOH, 20:1); mp 172–173.5 °C; ^1^H NMR (400 MHz, DMSO-d_6_) δ 7.12 (d, J = 9.0 Hz, 2H, Ar), 6.97 (d, J = 9.0 Hz, 2H, Ar), 5.53 (s, br, 2H, exch., 2-NH_2_), 3.51 (s, 3H, N_4_-CH_3_), 2.39 (s, 3H, -CH_3_), 2.68–2.64 (m, 2H, -CH_2_-), 1.67–1.59 (m, 4H, -CH_2_CH_2_-), 1.48–1.44 (m, 2H, -CH_2_). Anal. Calcd. for C_18_H_20_N_4_S_2_ 0.18 CH_3_OH: C, 60.26; H, 5.77; N, 15.46; S, 17.69. Found: C, 60.24; H, 5.82; N, 15.47; S, 17.72.

4-(6-Methoxy-3,4-dihydroquinolin-1(2H)-yl)-5,6,7,8-tetrahydrobenzo[4,5]thieno[2,3-*d*]- pyrimidin-2-amine (**6**): To a solution of **20** (100 mg, 0.417 mmol) in isopropanol (8 mL), 1–2 drops of HCl were added, followed by addition of 6-methoxy-1,2,3,4-tetrahydroquinoline (75.0 mg, 0.458 mmol), and the mixture was kept under reflux. The reaction mixture was cooled to room temperature, silica gel (250 mg) was added, and the solvent was removed under reduced pressure. Purification was performed by column chromatography using 1% MeOH in CHCl_3_ as the eluant, and the fractions containing the product (TLC) were pooled. The solvent was evaporated to give an off-white solid that was then washed with CHCl_3_ to afford 81.48 mg (48% yield) of **6**. TLC Rf = 0.32 (CHCl_3_: MeOH, 20:1); mp 190–191.8 °C; ^1^H NMR (500 MHz, DMSO-d_6_) δ 6.79 (d, J = 2.9 Hz, 1H, Ar), 6.56 (dd, J = 8.9, 3.0 Hz, 1H, Ar), 6.39 (d, J = 8.8 Hz, 1H, Ar), 5.18 (s, br, 2H, exch., 2-NH_2_), 3.95–3.88 (m, 2H, -CH_2_), 3.82 (s, 3H, -OCH_3_), 2.75 (t, J = 6.5 Hz, 2H, -CH_2_), 2.68–2.64 (m, 2H, -CH_2_), 1.94–188 (m, 2H, -CH_2_), 1.68–1.61 (m, 4H, -CH_2_CH_2_), 1.49–1.45 (m, 2H, -CH_2_). HRMS (ESI) calculated for C_20_H_22_N_4_OS [M+H]^+^, 367.48. Found: 366.80. HPLC analysis: retention time, 13.63 min; peak area, 95.51%; eluent A, H_2_O: eluent B, ACN; gradient elution (100% H_2_O to 10% H_2_O) over 60 min with flow rate of 0.5 mL/min and detection at 245 nm; column temperature, room temperature.

*N*-(4-methoxyphenyl)-*N*-methyl-5,6,7,8-tetrahydrobenzo[4,5]thieno[2,3-*d*]pyrimidin-4-amine (**7**): To a solution of **22** (200 mg, 0.890 mmol) in isopropanol (10 mL), 1–2 drops of HCl were added, followed by addition of 4-methoxy-*N*-methylaniline (134.0 mg, 0.979 mmol), and the mixture was kept under reflux for 6 h. The reaction mixture was cooled to room temperature, silica gel (400 mg) was added, and the solvent was removed under reduced pressure. Purification was performed by column chromatography using hexane and ethyl acetate (10:1) to give 98.50 mg (68% yield) of **7** as a white solid. TLC Rf = 0.48 (hexane: EtOAc, 3:1); mp 186.0–188.0 °C; ^1^H NMR (400 MHz, DMSO-d_6_) δ 8.58 (s, 1H, Ar), 6.93 (d, J = 9.1 Hz, 2H, Ar), 6.86 (d, J = 9.1 Hz, 2H, Ar), 3.71 (s, 3H, -OCH_3_), 3.42 (s, 3H, N_4_-CH_3_), 2.72 (t, 2H, -CH_2_), 1.74 (t, J = 6.1 Hz, 2H, -CH_2_), 1.61–1.50 (m, 2H, -CH_2_), 1.46–1.35 (m, 2H, -CH_2_). Anal. Calcd. for C_18_H_19_N_3_OS: C, 66.43; H, 5.88; N, 12.91; S, 9.85. Found: C, 66.72; H, 5.79; N, 12.88; S, 9.66.

*N*-(4-methoxyphenyl)-2-methyl-5,6,7,8-tetrahydrobenzo[4,5]thieno[2,3-*d*]pyrimidin-4-amine (**9**): Compound **22** (250 mg, 1.05 mmol) was dissolved in isopropanol (10 mL), followed by the addition of 1–2 drops of HCl and reacted with p-anisidine (0.15 mL, 1.36 mmol) for 4 h at reflux. The reaction mixture was cooled, and silica gel was added to the solvent mixture to prepare a silica gel plug. A flash column chromatographic separation was performed using ethyl acetate-hexane as eluent to afford intermediate **9** as a white solid (170.0 mg, 50% yield); TLC Rf = 0.22 (hexane: EtOAc, 3:1). mp, 208.4–210.1 °C; ^1^H NMR (400 MHz, DMSO-d_6_) δ 8.04 (s, 1H, br, exch., NH), 7.71 (d, J = 8.8 Hz, 2H), 7.31 (d, J = 8.8 Hz, 2H), 3.82 (s, 3H, -OCH_3_), 3.16–3.08 (m, 2H, -CH_2_), 2.85–2.76 (m, 2H, -CH_2_), 2.48 (s, 3H, -CH_3_), 1.88–1.78 (m, 4H, -CH_2_). HRMS (ESI) calculated for C_18_H_19_N_3_OS [M+H]^+^, 326.12. Found: 326.08. HPLC analysis: retention time, 10.10 min; peak area, 95.10%; eluent A, H_2_O: eluent B, ACN; gradient elution (100% H_2_O to 10% H_2_O) over 60 min with flow rate of 0.5 mL/min and detection at 240 nm; column temperature, room temperature.

*N*-(4-methoxyphenyl)-*N*,2-dimethyl-5,6,7,8-tetrahydrobenzo[4,5]thieno[2,3-*d*]pyrimidin-4-amine (**8**): Compound **9** (150 mg, 0.460 mmol) was dissolved in DMF (10 mL), followed by portion-wise addition of sodium hydride (60% in mineral oil) (18.44 mg, 0.460 mmol). Iodomethane (0.29 mL, 0.460 mmol) dissolved in 5 mL of DMF was added drop-wise to the suspension. The reaction was stirred for 2 h at room temperature. Silica gel was added to the solvent mixture and a plug was prepared. A flash column chromatographic separation was performed using ethyl acetate hexane as eluent to afford **8** as a white solid (106.0 mg, 68% yield); TLC Rf = 0.41 (hexane: EtOAc, 3:1); mp, 198–199.6 °C; ^1^H NMR (400 MHz, DMSO-d_6_) δ 6.92 (d, J = 9.1 Hz, 2H, Ar), 6.86 (d, J = 9.1 Hz, 2H, Ar), 3.72 (s, 3H, -OCH_3_), 3.42 (s, 3H, N_4_-CH_3_), 2.70 (t, J = 6.4 Hz, 2H,-CH_2_), 2.56 (s, 3H, 2-CH_3_), 1.72 (t, J = 6.0 Hz, 2H, -CH_2_), 1.60–1.52 (m, 2H, -CH_2_), 1.46–1.34 (m, 2H, -CH_2_). Anal. Calcd. for C_19_H_21_N_3_OS: C, 67.22; H, 6.24; N, 12.37; S, 9.45. Found: C, 67.38; H, 6.23; N, 12.31; S, 9.44.

2-Methyl-*N*-(4-(methylthio)phenyl)-5,6,7,8-tetrahydrobenzo[4,5]thieno[2,3-*d*]pyrimidin-4-amine (**11**): Compound **22** (300 mg, 1.26 mmol) was dissolved in toluene (10 mL), followed by addition of 1–2 drops of HCl and reacted with 4-(methylthio)aniline (227.43 mg, 1.63 mmol) for 6 h at reflux. The reaction mixture was cooled, and silica gel was added to the solvent mixture with the preparation of a plug. A flash column chromatographic separation was performed using ethyl acetate-hexane as eluent to afford intermediate **11** as a pale yellow solid (266.0 mg, 62% yield); TLC Rf = 0.15 (hexane: EtOAc, 3:1). mp, 201–202 °C; ^1^H NMR (400 MHz, DMSO-d_6_) δ 8.02 (s, 1H, br, exch., NH), 7.67 (d, J = 8.7 Hz, 2H, Ar), 7.27 (d, J = 8.7 Hz, 2H, Ar), 3.14–3.05 (m, 2H, -CH_2_), 2.83–2.77 (m, 2H, -CH_2_), 2.48 (s, 3H, -CH_3_), 2.45 (s, 3H, -CH_3_), 1.89–1.79 (m, 4H, -CH_2_). Anal. Calcd. for C_18_H_19_N_3_S_2_ 0.17 C_6_H_5_CH_3_ 0.12 HCl: C, 63.76; H, 5.71; N, 11.63; S, 17.75. Found: C, 63.77; H, 5.56; N, 11.76; S, 17.65.

*N*,2-dimethyl-*N*-(4-(methylthio)phenyl)-5,6,7,8-tetrahydrobenzo[4,5]thieno[2,3-*d*]pyrimidin-4-amine (**10**): Compound **11** (200 mg, 0.585 mmol) without further characterization was dissolved in DMF (10 mL), followed by portion-wise addition of sodium hydride (60% in mineral oil) (23.42 mg, 0.585 mmol). To the suspension was added drop-wise iodomethane (0.37 mL, 0.585 mmol) dissolved in 5 mL of DMF. The reaction was stirred for 2 h at room temperature. Silica gel was added to the solvent mixture, and a plug was prepared. A flash column chromatographic separation was performed using ethyl acetate hexane as eluent to afford **10** as a yellow solid (145.0 mg, 70% yield); TLC Rf = 0.45 (hexane: EtOAc, 3:1); mp, 189–191 °C; ^1^H NMR (500 MHz, DMSO-d_6_) δ 7.16 (d, J = 8.8 Hz, 2H, Ar), 6.85 (d, J = 8.8 Hz, 2H, Ar), 3.41 (s, 3H, N_4_-CH_3_), 2.69 (t, J = 6.2 Hz, 2H, -CH_2_), 2.55 (s, 3H, 2-CH_3_), 2.41 (s, 3H, -SCH_3_), 1.83 (t, J = 6.0 Hz, 2H, -CH_2_), 1.62–1.50 (m, 2H, -CH_2_), 1.46–1.37 (m, 2H, -CH_2_). Anal. Calcd. for C_19_H_21_N_3_S_2_: C, 64.19; H, 5.95; N, 11.82; S, 18.04. Found: C, 64.44; H, 5.99; N, 11.70; S, 17.78.

4-(6-Methoxy-3,4-dihydroquinolin-1(2H)-yl)-2-methyl-5,6,7,8-tetrahydrobenzo[4,5]thieno[2,3-*d*]pyrimidine (**12**): Compound **22** (250 mg, 1.05 mmol) was dissolved in isopropanol, followed by addition of 1–2 drops of HCl and 6-methoxy-1,2,3,4-tetrahydroquinoline (188 mg, 1.15 mmol). The reaction mixture was stirred for 8 h at reflux. The reaction mixture was cooled, and silica gel was added to the solvent mixture with the preparation of a plug. A flash column chromatographic separation was performed using hexane and ethyl acetate as eluent to afford **12** as a yellow solid (210 mg, 55% yield); TLC Rf = 0.53 (hexane: EtOAc, 3:1); mp, 199.0–201.0 °C; ^1^H NMR (400 MHz, DMSO-d_6_) δ 6.78 (d, J = 3.1 Hz, 1H, Ar), 6.54 (dd, J = 8.8 Hz, J = 3.1 Hz, 1H, Ar), 6.34 (d, J = 8.8 Hz, 1H, Ar), 3.80–3.71 (m, 2H, -CH_2_), 3.69 (s, 3H, -OCH_3_), 3.48–3.42 (m, 2H, -CH_2_), 2.82–2.70 (m, 4H, -CH_2_), 2.54 (s, 3H, 2-CH_3_), 1.99–1.92 (m, 2H, -CH_2_), 1.68–1.58 (m, 2H, -CH_2_), 1.51–1.40 (m, 2H, -CH_2_). Anal. Calcd. for C_21_H_23_N_3_OS 0.39 (CH_3_)_2_CHOH: C, 68.46; H, 6.77; N, 10.79; S, 8.24. Found: C, 68.78; H, 6.39; N, 10.52; S, 7.91.

6-((2-Methyl-5,6,7,8-tetrahydrobenzo[4,5]thieno [2,3-*d*]pyrimidin-4-yl)amino)naphthalen-1-ol (**14**): Compound **22** (300 mg, 1.26 mmol) was dissolved in isopropanol, followed by addition of 1–2 drops of HCl and reacted with 6-aminonaphthalen-1-ol (220 mg, 1.38 mmol) for 6 h at reflux. The reaction mixture was cooled, and silica gel was added to the solvent mixture to prepare a plug. A flash column chromatographic separation was performed using ethyl acetate-hexane as eluent to afford intermediate **14** as a pale-yellow solid (295.0 mg, 0.892 mmol, 65% yield); TLC Rf = 0.21 (hexane: EtOAc, 3:1); mp, 214.2–215.8 °C. ^1^H NMR (400 MHz, DMSO-d_6_) δ 8.56 (s, br, 1H, exch., -OH), 7.85 (d, J = 9.2 Hz, 1H, Ar), 7.62 (d, J = 7.5 Hz, 1H, Ar), 7.46 (s, br, 1H, exch., -NH), 7.35–7.22 (m, 2H, Ar), 6.76 (dd, J = 9.2, 2.4 Hz, 1H, Ar), 6.10 (d, J = 7.5 Hz, 1H, Ar), 2.65 (t, 2H, -CH_2_), 2.58 (s, 3H, 2-CH_3_), 1.89 (t, 2H, -CH_2_), 1.58–1.51 (m, 2H, -CH_2_), 1.45–1.38 (m, 2H, -CH_2_).Anal. Calcd. for C_21_H_19_N_3_OS 0.19 (CH_3_)_2_CHOH 0.13 HCl: C, 68.59; H, 5.51; N, 11.13; S, 8.49. Found: C, 68.78; H, 5.30; N, 11.63; S, 8.87.

*N*-(5-methoxynaphthalen-2-yl)-*N*,2-dimethyl-5,6,7,8-tetrahydrobenzo[4,5]thieno[2,3-*d*]pyrimidin-4-amine (**13**): Compound **14** (250 mg, 0.691 mmol) was dissolved in DMF (10 mL), followed by portion-wise addition of sodium hydride (60% in mineral oil) (28.0 mg, 0.691 mmol). Iodomethane (0.043 mL, 0.691 mmol) dissolved in 5 mL of DMF was added drop-wise to the suspension. The reaction was stirred for 2 h at room temperature. Silica gel was added to the solvent mixture, and a plug was prepared. A flash column chromatographic separation was performed using ethyl acetate-hexane as eluent to afford **13** as a yellow solid (153.56 mg, 57% yield); TLC Rf = 0.46 (hexane: EtOAc, 3:1); mp, 205.1–207.2 °C; ^1^H NMR (400 MHz, DMSO-d_6_) δ 7.99 (d, J = 9.2 Hz, 1H, Ar), 7.32 (d, J = 7.8 Hz, 1H, Ar), 7.29–7.24 (m, 2H, Ar), 7.09 (dd, J = 9.1, 2.4 Hz, 1H, Ar), 6.82 (d, J = 7.5 Hz, 1H, Ar), 3.92 (s, 3H, -OCH_3_), 3.53 (s, 3H, N_4_-CH_3_), 2.69 (t, 2H, -CH_2_), 2.61 (s, 3H, 2-CH_3_), 1.87 (t, 2H, -CH_2_), 1.51–1.45 (m, 2H, -CH_2_), 1.40–1.32 (m, 2H, -CH_2_). Anal. Calcd. for C_23_H_23_N_3_OS: C, 70.92; H, 5.95; N, 10.78; S, 8.23. Found: C, 70.75; H, 6.11; N, 10.53; S, 7.95.

### 3.2. Molecular Modeling

Docking of target compounds **4**–**14** was carried out in the colchicine site of tubulin (PDB: 6BS2, 2.65 Å). The crystal structure PDBs were obtained from the protein database. All docking procedures were performed using various modules of the Schrödinger Maestro suite (Schrödinger, LLC, New York, NY, USA, 2020–2) [49]. The protein was optimized and prepared for docking using the Maestro Protein Preparation Wizard to assess bond order and add missing hydrogens, followed by energy minimization using the OPLS3e force field. Gaps in the protein structures were ignored, as they were far from the active site. The Maestro Induced-fit Grid Generation module was then used to define a 15 × 15 × 15 Å grid from the center of all the ligands. Ligands used in the computational docking study were built using the Maestro 2D Build module. The Maestro LigPrep module was then used to generate conformers of each compound subjected to energy minimization using the OPLS3e force field protocol. The resulting compounds were docked into the prepared protein using the Maestro Induced Fit Docking. Induced Fit Docking was performed with standard precision with flexible ligand sampling. A total of 20 initial poses were generated for each compound. Based on the pose score, the top 4 poses were selected and subjected to energy minimization using the OPLS3e force field. Finally, the top 2 poses per compound were generated and ranked according to the Glide score, which is an approximation of binding energy defined by receptor–ligand complex energies. The top pose was analyzed and presented in the Biological Evaluation and Discussion section. Docking scores are listed in Appendix A (Appendix A).

### 3.3. Biological Studies

#### 3.3.1. Effects of Compounds on Cellular Microtubules

The effects of the compounds on cellular microtubules were evaluated in A-10 cells using indirect immunofluorescence microscopy. These cells were obtained from the American Type Culture Collection (ATCC) (Manassas, VA, USA). Cells were treated with the compounds of interest for 18 h, and the cells fixed with cold MeOH and microtubule structures were visualized using a β-tubulin antibody (Sigma-Aldrich, St. Louis, MO, USA). The concentration that caused loss of 50% of the interphase microtubules was defined as the EC_50_ and calculated as previously described [52]. These values represent an average of at least three independent experiments.

#### 3.3.2. Sulforhodamine B (SRB) Assay

The antiproliferative and cytotoxic effects of the compounds in cancer cells were evaluated using the SRB assay [53] as previously described [54]. MDA-MB-435 cells were obtained from the Lombardi Cancer Center of Georgetown University (Washington, DC, USA). SK-OV-3 and HeLa cells were purchased from ATCC. Details about the generation of the SK-OV-3 MDR1-M6/6 and HeLa WTβIII cells were described previously [54]. The IC_50_ values represent an average of three independent experiments, each conducted using triplicate points.

#### 3.3.3. Quantitative Tubulin Studies

Bovine brain tubulin was purified as described previously [55]. The tubulin assembly assay has been described in detail [50]. Briefly, 1.0 mg/mL of tubulin (10 µM) was preincubated for 15 min at 30 °C in 0.8 M monosodium glutamate (pH of 2 M stock solution adjusted to 6.6 with HCl), varying compound concentrations and 4% (*v*/*v*) DMSO as compound solvent. After the preincubation, the reaction mixtures were placed on ice and 0.4 mM GTP was added. The reaction mixtures were transferred to cuvettes at 0 °C in a recording spectrophotometer equipped with an electronic temperature controller. After baselines were established, the temperature was elevated over about 30 s to 30 °C, and changes in turbidity were monitored at 350 nm for 20 min. The compound concentration that caused a 50% reduction in increase in turbidity, interpolated from the values obtained with defined compound concentrations, was defined as the IC_50_ value. The assay to measure inhibition of [^3^H]colchicine binding was described in detail previously [56]. Briefly, 0.1 mg/mL (1.0 µM) tubulin was incubated at 37 °C with 5.0 µM [3H]colchicine and potential inhibitors at 5.0 µM. Incubation was for 10 min, at which point the reaction had reached 40–60% of the maximum colchicine that can be bound in reaction mixtures without inhibitor. The [^3^H]colchicine was a product of PerkinElmer, USA. CA-4 was a generous gift of Dr. G. R. Pettit, Arizona State University.

#### 3.3.4. Cell Culture

HeLa, A-10, SK-OV-3, and SK-OV-3 MDR1-M6/6 cells were grown in Basal Medium Eagle (Sigma-Aldrich, St. Louis, MO, USA) supplemented with 10% FBS (Hyclone, GE Life Sciences, Logan, UT, USA) 1% GlutaMAX (Gibco, Life Technologies, Waltham, MA, USA) and 50 µg/mL gentamycin (Life Technologies, USA). HeLa WTβIII cells were grown in Dulbecco’s Modified Eagle Medium (Life Technologies) supplemented with 10% FBS and 50 µg/mL gentamycin. MDA-MB-435 cells were maintained in Improved Minimum Essential Medium (Life Technologies) supplemented with 10% FBS and 25 µg/mL gentamycin. All cells were grown at 37 °C in a humidified environment with 5% CO_2_.

#### 3.3.5. MDA-MB-435 Xenograft Model

MDA-MB-435 tumor fragments were implanted s.c. into the flanks of female nude mice. Once tumors reached ~200 mm^3^, mice were injected i.p. with compound **4** (75 mg/kg) or paclitaxel (15 mg/kg) 3 times a week. Compound **4** was dissolved in a 50:50 Cremophor EL/DMSO mixture and further dissolved in PBS for a final concentration of 10:10:80 Cremophor EL:DMSO:PBS (*v*/*v*). Paclitaxel was dissolved in a 50:50 Cremophor EL/ EtOH mixture and further dissolved in PBS for a final solvent concentration of less than 5% (*v*/*v*) Cremophor/EtOH in PBS. Tumor volumes and mouse weights were measured 2–3 times a week. A two-way ANOVA with Dunnett’s post-hoc tests including all treatment groups in the trial was used to determine significant differences in tumor volumes between drug-treated and untreated control groups on each measurement day. These animal studies were performed at the University of Texas Health Science Center in San Antonio in compliance with an approved Institutional Animal Care and Use protocol.

## 4. Conclusions

A series of 11 novel 5,6,7,8-tetrahydrobenzo[4,5]thieno[2,3-*d*]pyrimidines were designed, synthesized and assessed as MTAs. Several of these analogues were significantly more potent than the lead compounds and circumvented drug resistance mediated by Pgp and βIII-tubulin. Compound **4** showed statistically significant antitumor effects in an in vivo xenograft model (MDA-MB-435). This study corroborated our molecular modeling predictions, which suggested structural variations to improve binding at the CS to afford better MTAs for the potential treatment of cancer. Compound **4** is a candidate for further evaluation in preclinical studies to evaluate its antitumor efficacy.

## Data Availability

Not applicable.

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
