# Peer review of "Design, Synthesis, and Biological Evaluation of 5,6,7,8-Tetrahydrobenzo[4,5]thieno[2,3-d]pyrimidines as Microtubule Targeting Agents"

_molecules, 2022, doi:10.3390/molecules27010321_

Round 1
Reviewer 1 Report
Reviewer Comments to Author
Current research manuscript describes Design, Synthesis, and Biological Evaluation of 5,6,7,8-tetrahy-drobenzo[4,5]thieno[2,3-d]pyrimidines as Microtubule Targeting Agents. To the best of reviewer’s understanding, current version of this article unfolds an additional avenue of substitution primarily on pyrimidine core and adds a new finding to the existing literature to display interesting and useful characteristics. Herein, I would like to recommend accepting current version of this manuscript for publication in molecules but after major revision. To this end, there are some suggestions related to the manuscript and authors should take into account these suggestions before submitting a revised version.
- It is suggested to add EC50 values of compound 1, 2 and 3 against A-10 cells and MDA- 104 MB-435 in Figure 2.
- Lines: 110-114. Are these lines ok and explains what is given in table1? Reviewer feels that only compounds 5,10, and 11 were only substituted with S atom at not 4-14. This should be modified if not clear.
- Scheme 4: The chemical structures of compounds 9, 11, and 14 are similar to compounds 8,10, 13. There should be some modification.
- Author sates that “compounds 9, 11, and 14 did not show any activity in the microtubule depolymerization assay, and these were not evaluated for antiproliferative effects, corroborating our previous reports that the N4-Me is crucial for MT activity”. This makes sense. However, reviewer feels that despite of huge efforts to find a potent compound like 4 which is substituted with 4´-MeO. Why author not surveyed 4´-OEt or 4´-O-iPr substitutions to eliminate effects of alkyl chain and to establish the suitability of 4-MeO only in the series under study? Reference 38 (doi: 10.1016/j.bmc.2012.12.010) clearly indicates that 4´-OEt analog exhibited (IC50 ± SD (nM) (MDA-MB-435= 14.4 ± 0.5, EC50 for microtubule depolymerization in A-10 cells (nM)=83 ± 4 and EC50/IC50 =6.5). What was the driving force to select only 4´-MeO moiety and non-inclusion of 4´-OEt analogs?
- The author has added Table 5 which demonstrates the human cancer cell growth inhibitory activity GI50 (nM) of 4 in the NCI-60 Cell Line Panel, but no detail has been provided in text. A brief description regarding Table 5 should be added wrt compound 4 to benefit the readership of journal.
- Did author perform TLC in hexane or hexanes. Please rectify accordingly.
- Though description regarding structural elucidation is given in section 3.1 however, in all cases 13CNMR are missing. Reviewer would like to suggest adding full set of labelled 1HNMR and 13CNMR spectra as SI file for 1)-final analogs, and 2)-all unknown analogs synthesized in this study.
Author Response
Dear Editor and Reviewer 1,
We have provided a point-by-point response to the reviewer’s comments and uploaded it as a Word file. Please see the attachment.
Best
Dr. Aleem Gangjee

Reviewer 2 Report
I have reviewed the manuscript submitted by Susan Mooberry, Aleem Gangjee and co-workers, which deals of the design, synthesis and biological evaluation of some nitrogen-and sulfur-containing heterotricycles.
This work is well conducted and could be of interest for people working on the design of Microtubule Binding Agents (MBA).
Hence, I do recommend the publication of this work in Molecules, after some major corrections.
Authors provide in this article an extended investigation of a previous work dedicated to the design of MBA having cyclopenta[d]pyrimidines (reference 36: Bioorg Med Chem 2021, 29, 115887).
While they also mention the isoelectronic replacement of the NH in analogues 1-3 by a sulphur atom and the reduction of the aromatic ring, the described molecules in this paper could also be seen as a ring addition/modification, from cyclopentyl (ref 36) to tetrahydrobenzothiophenyl moiety. This similarity should be indicated at the beginning of the manuscript, in Figure 2.
If this could be seen as a routine extension of a previously described approach, their findings could be of interest to better understand the structural modifications of 5,6,7,8-tetrahy-2 drobenzo[4,5]thieno[2,3-d]pyrimidines and their effects on activity.
Concerning this point, I strongly recommend authors to provide in this article an exhaustive bibliography on 5,6,7,8-tetrahy-2 drobenzo[4,5]thieno[2,3-d]pyrimidines that are already described in the literature, and especially those developed as anticancer agents. Indeed, a fast database structure-oriented search gives some interesting results: EGFR/HER2 inhibitors (Bioorganic Chemistry 88 (2019) 102944 / European Journal of Medicinal Chemistry 2010, 45, 2473-2479), treatment of hyperproliferative disorders (PCT Int. Appl WO2005010008 (A1)), antitumor agents (GANGJEE Aleem – PCT Int. Appl. WO2010/6032, 2010, A1) … This bibliographic search must be done properly.
In the abstract, please indicate which structure (tetrahydrobenzothiophene) is prepared according to Gewald reaction and also a brief comment on the structural requirements of the most active compounds 4,5 and 7.
The introducing part gives a complete and detailed insight on MTAs, focusing on the last part of Colchicine Site CS destabilizers.
I think that the location of Table 1 should be after Schemes 1-4, describing the chemical syntheses of compounds 4-14.
There is, to my opinion, no sense to indicate which molecules were prepared before Figure 3 and Scheme 1-4.
To be clearer with this manuscript new organization, the envisaged structural modifications must be indicated in Figure 2, by using circles/squares onto compounds 1-3 and the envisaged molecular targets. The term “Lead” must be removed as the new series (compounds 4-14) is not “improved” from the lead compounds 1-3 (a lead compound has, by definition, improved ADME properties than the hits…). Please also remove “lead” in the following paragraphs.
The abbreviation “Ar” in the third column of Table 1 is not correct as these groups are “R2-N-Ar” groups (and so, R should be R1). So, please correct accordingly (and also in Schemes 3,4…).
First column should be “Compound N°” instead of “No”.
The rationale of the drug design is well explained in part 2.1 and mentioned isosteric replacement of the NH to S atom. Please mention literature referenceS for which such a strategy was observed for MTAs/MBAs. You will find some examples in the literature of biological activity improvement by switching NH for a S atom…
Instead of “increasing numbers of sp3 fragments” authors should mention “decrease numbers of sp2 bonds” as these bonds are responsible of the “flatness” of MTAs…
I know that this is discussed later but a precision of what type of amino acids are involved in “hydrogen bond interactions with corresponding amino acids at the CS.”
The conformational restrictions are properly depicted and bring an interesting insight.
In the molecular modelling section 2.2, please indicate that the used pdb file 6BS2 is a co-crystal structure with a pyrimidine derivative (indicate its structure somewhere…). A comparison of the co-crystalized pyrimidine derivative (ref 31) would be really informative, by comparison with compound 4 and colchicine. There are also some recent literature precedents describing the differences in molecular modelling studies, depending on the co-crystalized/docked compounds (eg comparisons between docking results using different two different crystal structures, obtained with CA-4/ colchicine co-crystals).
The syntheses of the target compounds 4-14 are appropriately described: some minor corrections have to be done: indicate the yield in Scheme 1 for intermediates 20-22.
Scheme 2: simplify the Boc group in 24,25.
Scheme 3: indicate which compounds (4-7, 11, 12, 14) are obtained starting from 20, 21 or 22, with the corresponding yields. Scheme 4: same remark…
In section 2.4.1, the sentence “Motivated by the promising docking results described above” sounds odd because I guess the docking studies were realized a posteriori to the biological evaluations… Hence, I suggest the authors to remove this sentence and to place the docking studies after the biological evaluations’ results…
I do not understand why authors mention “the concentration that causes the loss of 50% of cellular microtubules as visualized microscopically” as it is specified in the experimental part that the tubulin polymerization test relied on turbidimetry (as many papers in the literature…). Please correct this sentence.
The tubulin assembly inhibition properties of compounds 4-14 described in the present paper were compared with literature references (1-3 and CA-4) and presented valuable activities, binding on the colchicine site of tubulin.
For antiproliferative activities, the comparison was also done with the microtubule-stabilizing compound paclitaxel, in complement with microtubule-destabilizing agent CA-4. The effect of the designed compounds proved to be interesting as anti-proliferative agents, including on Pgp-mediated MDR cancer cells, with favourable resistance ratios Rr (higher than CA-4 and much higher than paclitaxel).
Nonetheless, the most active compound 4, which was then evaluated on the NCI-60 cell line panel, remains less active than the known reference CA-4 on HeLa and SK-OV-3 cells (except for resistant cells treated with paclitaxel). This last observation is in contradiction with the tubulin assembly inhibitions observed in Table 3, where compounds 4, 5, 7, 8 and 10 were found to be more active than CA-4. Authors did not give any plausible explanations on this fact: to shed some light on this fact, authors must propose such an explanation.
The monitoring of cell cycle, by using flow cytometry, should be carried out, to determine which cycle transition is blocked. While not mandatory for this paper, authors should have provided these data.
Authors have finally carried out a in vivo study on MDA-MB-435 xenografted mice: this study have shown that the most promising compound 4 at 75 mg/kg was more efficient than paclitaxel at 15 mg/kg.
Authors must detail the following sentence “In this trial the effects of paclitaxel (15 mg/kg) were not significantly different than control at day 14.” because on Figure 5, this difference seems to be there (tumour volume for control in black line is much higher than TTX in red line…). Could you bring some more detailed explanations?
In Table 2, I strongly suggest the authors to remove the last column “EC50/IC50 ratio” as this ratio compares two different effects: antiproliferative effects and microtubule depolymerisation abilities. So, there is no sense to compare such values.
A MAJOR correction MUST be done for this paper as neither 13C NMR spectra, nor MS data, for the described compounds, were provided. These data MUST be provided to assess the identity of the newly prepared compounds.
In the conclusion, authors MUST correct the following sentence “Several of these analogues were significantly more potent than the lead compounds and CA-4 and circumvented drug resistance mediated by Pgp and βIII-tubulin.”, as it was demonstrated in the paper that compound 4 was LESS active than CA-4 when considering the anti-proliferative activities. Please correct by indicating that most of the obtained compounds were more active than CA-4 for tubulin polymerization inhibition only!
Similarly, I do not agree when the authors claimed that “This study corroborated our molecular modeling predictions that suggested structural variations to improve binding at the CS to afford better MTAs for the potential treatment of cancer”.
To assess such a thing (in the conclusion but a similar sentence was also used in the abstract…), a detailed comparison must have been carried out with CA-4, the reference used during in vitro biological evaluations. I have the feeling that the molecular modelling study must be investigated more deeply.
There are some typo mistakes. In address list, e-mail@e-mail.com Table 4, “M66”
There are also English language mistakes: in the keywords, “targetting". Correct “in A-10 cells” in “with EC50 values of 130, 103 1100 and 1200 nM in A-10 cells” “was THE most potent compound”
The supplementary file does not contain much information: this file should contain 1H and 13C NMR spectra of synthesized compounds. So, these spectra should be provided, for an appropriate information of the reader.
Author Response
Dear Editor and Reviewer 2,
We have provided a point-by-point response to the reviewer’s comments and uploaded it as a Word file. Please see the attachment.
Best
Dr. Aleem Gangjee

Round 2
Reviewer 1 Report
Dear Editor,
To the best of reviewer’s understanding, the authors have performed necessary modifications and corrections except comment No. 7 as under:
Past comment 7:
Though description regarding structural elucidation is given in section 3.1 however, in all cases 13CNMR are missing. Reviewer would like to suggest adding full set of labelled 1HNMR and 13CNMR spectra as SI file for 1)-final analogs, and 2)-all unknown analogs synthesized in this study.
Author’s Reply:
We used elemental analysis and 1HNMR to characterize the final compounds. This is sufficient for the characterization of compounds as per the ACS guidelines.
Current Comment:
The study is being published in MDPI journal. As per MDPI policy for “Molecules” reports on previously undescribed organic compounds should include, as supplementary data, 1H, 13C and/or other key heteronuclear or 2D NMR spectra, together with High Resolution Mass Spectrometry (HRMS) or elemental analysis.
The reviewer did not ask for HRMS data but would like to recommend adding of all stuff (as per Journal policy) according to Comment No.7 (above) before the publication of this study.
Additional Correction: The added IC50 and EC50 values in Figure 2 should indicate which value belongs to which analog. This should be rectified too.
Sincerely,
Latif Mayo
Author Response
Dear Reviewer 1,
We appreciate your consideration of our revised manuscript and trust that we have adequately responded to the reviewer’s and editor’s comments and look forward to the acceptance of the manuscript.
Best
Dr. Aleem Gangjee
Reviewer 2 Report
I received the revised version of the manuscript and I have noticed that, even if many of my remarks were taken into consideration, some others were not.
Some of these remarks were critical (eg the complete characterization of novel compounds, including 13C NMR spectra) and MUST be considered before publication in Molecules.
Please find below some of these important remarks.
- REV#2: The envisaged structural modifications must be indicated in Figure 2, by using circles/squares onto compounds 1-3 and the envisaged molecular targets.
REV#2: The term “Lead” must be removed as the new series (compounds 4-14) is not “improved” from the lead compounds 1-3 (a lead compound has, by definition, improved ADME properties than the hits…). Please also remove “lead” in the following paragraphs.
- Addressed in line 178-180 and in the supplementary file in Figure S1.
REV#2: As this figure could be interesting for the reader, I strongly suggest to insert Figure S1 in the manuscript, rather than the supplementary information…
- Scheme 3: indicate which compounds (4-7, 11, 12, 14) are obtained starting from 20, 21 or 22, with the corresponding yields. Scheme 4: same remark…
Addressed in the experimental section.
REV#2: For clarity, this must be modified also in Scheme 3
- In section 2.4.1, the sentence “Motivated by the promising docking results described above” sounds odd because I guess the docking studies were realized a posteriori to the biological evaluations… Hence, I suggest the authors to remove this sentence and to place the docking studies after the biological evaluations’ results…
Addressed, and line has been removed.
REV#2: This was not addressed at all, as the docking studies is still before the biological evaluations’ results… So, this recommendation was not considered at all by the authors… So please do not mention it is addressed, as it is not…
- I do not understand why authors mention “the concentration that causes the loss of 50% of cellular microtubules as visualized microscopically” as it is specified in the experimental part that the tubulin polymerization test relied on turbidimetry (as many papers in the literature…). Please correct this sentence.
These two assays are different, one is a cell-based assay, evaluating loss of microtubules in cells and other is a biochemical assay using purified bovine brain tubulin. The advantage of the two different types of assays is that in cell-based assays, the compounds have to cross the plasma membrane to engage the target that has a full complement of accessory proteins that facilitate microtubule polymerization and depolymerization in cells and in the purified tubulin assay this is not the case, it is only tubulin, GTP and buffers, useful but not the same as a cellular system.
REV#2: As a contributor of the design and evaluation of new MBAs, I perfectly know what tubulin polymerization assays on isolated protein are or within the cell are (I personally published results using both approaches…). But there is a difference between the text and the experimental section. This should be clarified.
- In table 3, tubulin assembly inhibition was measured using purified tubulin, whereas the antiproliferative activities are cell-based assays. A correlation between a cell-based assay and a biochemical assay is not always observed. This might be due in part to the ability of the compounds to cross the cell membrane and accumulate intracellularly.
REV#2: I completely agree with the authors but this must be pointed out in the manuscript to avoid misunderstandings
- REV#2: The reviewer thanks the authors for these explanations. And I agree that a small comment must be added.
But I do not see in the revised manuscript that “a trend toward significant with paclitaxel, but it was not statistically significant.”
- REV#2: The reviewer thanks the authors for this explanation, which should also be (in a brief manner…) added in the main text.
- A MAJOR correction MUST be done for this paper as neither 13C NMR spectra, nor MS data, for the described compounds, were provided. These data MUST be provided to assess the identity of the newly prepared compounds.
We used elemental analysis and 1HNMR to characterize the final compounds. This is sufficient for the characterization of compounds as per the ACS guidelines.
REV#2: I published several ACS papers (several J. Med. Chem. articles) and I know these guidelines but the ACS guidelines mention “In select instances, a variety of definitive spectroscopic techniques may provide sufficient characterization (for example, if many of the nuclei are NMR active), but in the majority of cases evidence for elemental constitution must be provided by either elemental analysis (e.g. combustion analysis, microprobe analysis), or mass spectrometry.”
So, I maintain my demand (which was also required by reviewer #1…) to have the 13C NMR spectra for all the novel compounds. Molecules is a journal dedicated to new compounds, which requires a complete characterization. Hence, this is mandatory to provide a complete characterization of original compounds, including 13C NMR.
If the editorial office of Molecules would take the responsibility to publish novel compounds without 13C NMR spectra, please do so, but I think this is not serious…
- In the conclusion, authors MUST correct the following sentence “Several of these analogues were significantly more potent than the lead compounds and CA-4 and circumvented drug resistance mediated by Pgp and βIII-tubulin.”, as it was demonstrated in the paper that compound 4 was LESS active than CA-4 when considering the anti-proliferative activities. Please correct by indicating that most of the obtained compounds were more active than CA-4 for tubulin polymerization inhibition only!
Addressed
REV#2: I do not see any modification of the text… So that must not be written “addressed” and be corrected according to my comments…
- Similarly, I do not agree when the authors claimed that “This study corroborated our molecular modeling predictions that suggested structural variations to improve binding at the CS to afford better MTAs for the potential treatment of cancer”.
To assess such a thing (in the conclusion but a similar sentence was also used in the abstract…), a detailed comparison must have been carried out with CA-4, the reference used during in vitro biological evaluations. I have the feeling that the molecular modelling study must be investigated more deeply.
Lines removed from abstract. Supplementary data has been provided to rationalize the above statement.
REV#2: I guess there is a misunderstanding… This sentence “This study corroborated our molecular modeling predictions that suggested structural variations to improve binding at the CS to afford better MTAs for the potential treatment of cancer” was not located in the abstract but in the conclusion. And, in the revised version, it is still located in the conclusion.
So now, this sentence MUST be removed!
And I do not see many supplementary data that rationalize the above statement… Please take into consideration my remarks.
- The supplementary file does not contain much information: this file should contain 1H and 13C NMR spectra of synthesized compounds. So, these spectra should be provided, for an appropriate information of the reader.
We used elemental analysis and 1HNMR to characterize the final compounds. This is sufficient for the characterization of compounds as per the ACS guidelines.
REV#2: No, this is not sufficient, to my opinion! 13C NMR data MUST be provided!
Author Response
Dear Reviewer 2,
We appreciate your kind consideration of our revised manuscript and trust that we have adequately responded to the comments and look forward to the acceptance of the manuscript.
Thank you.
